# Sparse, Geometric Autoencoder Models of V1

**Jonathan Huml**                                    JHUML@G.HARVARD.EDU

**Abiy Tasissa**                                    ABIY.TASISSA@TUFTS.EDU

**Demba Ba**                                    DEMBA@SEAS.HARVARD.EDU

**Editors:** Sophia Sanborn, Christian Shewmake, Simone Azeglio, Arianna Di Bernardo, Nina Miolane

## Abstract

The classical sparse coding model represents visual stimuli as a convex combination of a handful of learned basis functions that are Gabor-like when trained on natural image data. However, the Gabor-like filters learned by classical sparse coding far overpredict well-tuned simple cell receptive field (SCRF) profiles. A number of subsequent models have either discarded the sparse dictionary learning framework entirely or have yet to take advantage of the surge in unrolled, neural dictionary learning architectures. A key missing theme of these updates is a stronger notion of *structured sparsity*. We propose an autoencoder architecture whose latent representations are implicitly, locally organized for spectral clustering, which begets artificial neurons better matched to observed primate data. The weighted-$\ell_1$ (WL) constraint in the autoencoder objective function maintains core ideas of the sparse coding framework, yet also offers a promising path to describe the differentiation of receptive fields in terms of a discriminative hierarchy in future work.

**Keywords:** Locality, Manifold Learning, Graph Laplacian, Phase Symmetry

## 1. Introduction

Overcomplete sparse coding as a model of the primary visual cortex (V1) is a pillar of computational neuroscience (Olshausen and Field, 1997). Training on natural image[1] patches via a Hebbian learning rule produces filters that are spatially localized, bandpass, and oriented to a select range of rotation angles. These filters are similar to those observed in the mammalian cortex (Jones and Palmer, 1987), which are well-described by two-dimensional Gabor functions. However, the properties of Gabor filters fitted to the simple cell receptive field (SCRF) estimates produced by sparse coding have been shown to misalign with filters fitted to rhesus macaque responses to drifting sinusoidal gratings (Ringach, 2002). In particular, the original sparse coding (SC) model overpredicts and underpredicts the number of well-tuned and broadly-tuned cells, respectively. Well-tuned cells maintain several (more elongated) subfields than the "blob-like" broadly tuned cells, as shown in Figure 1.

---

1. http://www.rctn.org/bruno/sparsenet/IMAGES.mat

A number of models have been subsequently proposed as a result. Of particular interest, (Rehn and Sommer, 2007) limits the number of active neurons rather than the average neural activity, which significantly improves diversity of shapes. (Zylberberg et al., 2011) develops a spiking network based on synaptically local information to overcome this discrepancy. Hypothesizing that explicit image reconstruction is not a biologically relevant task, (Yerxa and Simoncelli, 2022) proposes a novel contrastive objective, Local Low Dimensionality (LLD), that minimizes the dimensionality of encodings of spatially local image patches relative to their global dimensionality.

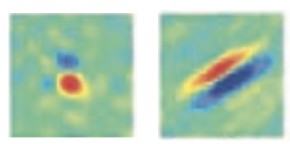

Figure 1: Broadly-tuned (left), well-tuned (right) macaque SCRFs Ringach (2002)

While LLD diversifies SCRF shapes compared to sparse coding, (Shen et al., 2019) uses deep methods to reconstruct the brain's perceptions of images from functional magnetic resonance imaging data, showing the relevance of generative models of the visual cortex. In addition to past success of the reconstructive framework, we therefore investigate a deep recurrent autoencoder architecture with additional regularization constraints to enforce a similar flavor of locality:

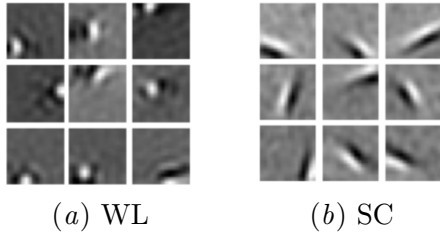

(a) WL    (b) SC

Figure 2: The weighted-$\ell_1$ penalty begets more of the "blob-like" SCRFs that are missing from the original model.

namely, a weighted-$\ell_1$ penalty (WL). While the hierarchical setting is left to future work by LLD, this architecture naturally learns a hierarchy of representational units when trained with an additional discriminative loss term (Rolfe and LeCunn, 2013). The reported findings motivate the need for *spatial regularization* of neurons and necessitate more precise arguments against reconstructive frameworks like sparse coding.

## 2. Previous Work

The general neural coding framework can be formulated as:

$$\mathcal{L}(\mathbf{A}, \mathbf{X}) = \frac{1}{2}||\mathbf{Y} - \mathbf{A}\mathbf{X}||_F^2 + S_\lambda(\mathbf{X}) \tag{1}$$

where $\mathbf{Y} \in \mathbb{R}^{d \times n}$ is a set of $n$ stimuli of dimension $d$, $\mathbf{A} \in \mathbb{R}^{d \times m}$ is a learned set of $m$ basis functions, $\mathbf{X} \in \mathbb{R}^{m \times n}$ is a set of $n$ latent representations of inputs, and $S_\lambda(\mathbf{X})$ is a regularization penalty. (Rozell et al., 2007), among other advances, associates sparse coding with $S_\lambda(\mathbf{X}) = ||\mathbf{X}||_1$ (columnwise). However, while neurons fire sparsely, they are also specialized to certain types of visual stimuli in the input space. As formulated, (1) makes no explicit assumptions about the structure of this sparsity in neural latent space.

LLD discards the reconstruction loss and encodes natural image stimuli to local ensembles of image patches $\{(\mathbf{x}_1^{(1)}, \ldots, \mathbf{x}_n^{(1)}), \ldots, (\mathbf{x}_1^{(B)}, \ldots, \mathbf{x}_n^{(B)})\}$, with superscipts denoting

local ensembles and subscripts denoting ensemble members. LLD is a shallow network where $\mathbf{s}_i^{(j)} = \text{ReLU}(\mathbf{W}\mathbf{x}_i^{(j)} + \mathbf{b})$. A covariance matrix $\mathbf{\Sigma}_l^{(j)} = \text{Cov}\left([\mathbf{s}_1^{(j)}, \ldots, \mathbf{s}_n^{(j)}]\right)$ is formed on the $j^{th}$ ensemble. The LLD loss is formulated as:

$$\mathcal{L}(\mathbf{W}, \mathbf{b}) = \frac{\mathbb{E}_j[\text{tr}(\mathbf{\Sigma}_l^{(j)})]}{\text{tr}\left(\mathbf{\Sigma}_g\right)} \tag{2}$$

where $\mathbf{\Sigma}_g$ is the response covariance to all patches in the batch. Due to the algebraic connection between trace and singular values, the numerator pushes the local ensembles to low dimensional subspaces, while the denominator pulls the ensembles to as many different low dimensional subspaces as possible. By exploiting this tradeoff between local and global subspaces, the LLD model is able to better replicate the diversity of SCRF shapes, which exhibit a phase symmetry in rhesus macaque data. These phases are obtained for each learned filter by fitting a two-dimensional Gabor function, and their distributions for the learned set of filters are shown in Figure 3.

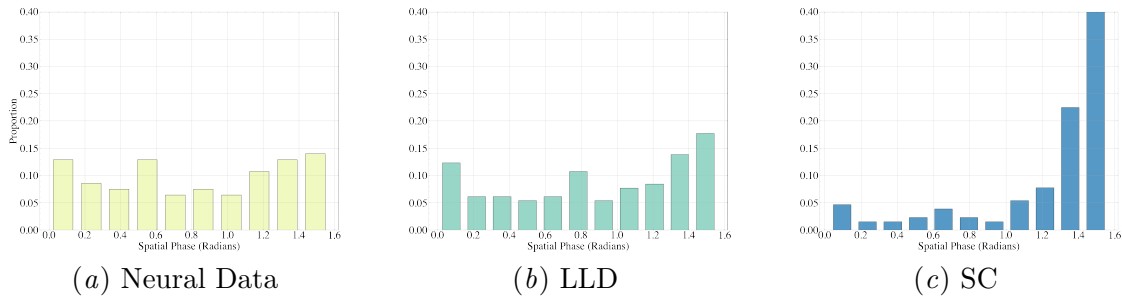

(a) Neural Data        (b) LLD        (c) SC

Figure 3: Gabor spatial phases of rhesus macaques are largely bimodal, but SC phases are highly skewed due to the absence of "blob-like" fields in Figures 1 and 2.

## 3. Locality-Constrained Reconstructive Frameworks

Can we incorporate additional structure into the reconstructive framework to better match the experimental data? The weighted-$\ell_1$ constraint penalizes neural encoding activity based on the distance between the natural image stimuli and the basis functions $\mathbf{a}_j$, where:

$$S_\lambda^{WL}(\mathbf{X}) = \mathbb{E}_{i \in [n]}\left[\sum_{j=1}^{m} x_j ||\mathbf{y}_i - \mathbf{a}_j||_2^2\right] \tag{3}$$

Here, neurons are specialized to certain types of input as large neural energy requirements will limit the strength of the firing rate $x_j$. In contrast to the original sparse coding alternating minimization scheme, we solve this through algorithm unrolling (Monga et al., 2020) into a deep recurrent autoencoder, which projects the encodings onto the probability simplex through a nonlinearity $\mathcal{P}_S$.

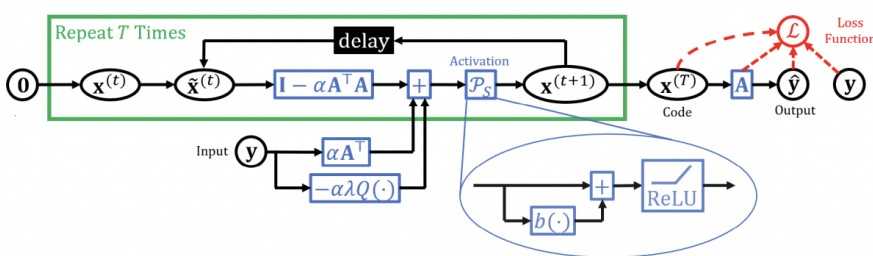

Figure 4: The unrolled architecture that learns $\mathbf{A}$ and $\mathbf{X}$, where $\mathcal{P}_S = \text{ReLU}\left(\mathbf{x} + b\left(\mathbf{x}\right) \cdot \mathbf{1}\right)$ and $Q\left(\mathbf{y}\right) = \sum_{j \in [m]} ||\mathbf{y} - \mathbf{a}_j||^2$ is a quadratic neuron. See appendix for details (Tasissa et al., 2021).

This penalty can be mathematically interpreted as a bipartite graph Laplacian on the $m + n$ basis functions and inputs (vertices), whose edge weights between the $y_i^{th}$ and $a_j^{th}$ vertices are $x_{ij}$, and 0 otherwise. Thus, the stimuli can be easily clustered by performing an eigendecomposition on this constructed Laplacian. In a discriminative classification task, this will allow for a more rigorous analysis of a given neuron's sensitivity to various class types.

We have also explored an iterative Laplacian scheme (Kodirov et al., 2015) where:

$$S_\lambda^{LAP}(\mathbf{X}) = \text{tr}(\mathbf{X}\mathcal{G}\mathbf{X^T}) \tag{4}$$

for a pre-constructed (or iteratively updated) graph Laplacian $\mathcal{G}$. The penalty (4) is typically used in addition to an $\ell_1$ penalty. Here, however, $\mathcal{G}$ is built on the stimuli space to preserve local pairwise distances in latent space, whereas the weighted-$\ell_1$ penalty essentially interpolates the manifold in $\mathbb{R}^d$ with the set $\mathbf{a}_{j \in [m]}$ and then uses as few basis functions as possible. Thus $S_\lambda^{LAP}$ only constrains firing rates, while $S_\lambda^{WL}$ constrains both the firing rates *and* the learned basis functions, which we refer to as "spatial regularization."

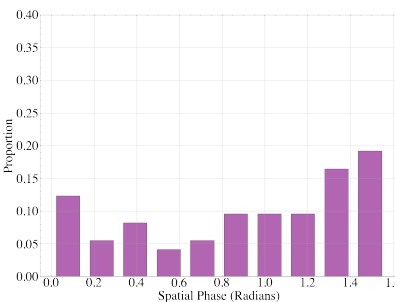

In our experiments, we find that the additional weighted-$\ell_1$ regularization technique shifts the spatial phase distribution to a more diverse range and vastly improves symmetry. The weighted-$\ell_1$ penalty makes the general sparse coding framework competitive with other local frameworks like LLD in terms of symmetry while maintaining the core ideas of the original model.

Figure 5: Spatial phases of the weighted-$\ell_1$ (WL) autoencoder. Locality-regularization is able to shift the original sparse code distribution of spatial phases

## 4. Conclusion

The improved spatial symmetry warrants further exploration into deep recurrent autoencoders (with varying flavors of locality constraints) as a model of the primary visual cortex. Is *explicit* image reconstruction biologically plausible? This assumption may be loosened in future work by considering a distribution of codes instead of a point estimate (Park and Pillow, 2020). However, given previous work showing the intrinsic hierarchical structure of discriminative recurrent sparse autoencoders (Rolfe and LeCunn, 2013), the findings presented here offer a potential path towards rigorously describing the differentiation of receptive fields that match experimental data.

## 5. Acknowledgements

Jonathan Huml would like to acknowledge support from the Harvard Institute for Applied Computational Science, and Demba Ba would like to acknowledge support from NSF National Science Foundation DMS-2134157.

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

## Appendix A. Unrolled Network and Training

The spatial phase plots are obtained through a three-step process:

1. Train the (unrolled) network on the original Sparsenet data (Olshausen and Field, 1997) image patches

2. Using the learned basis functions, obtain the simple cell receptive field estimates through spike-triggered averaging

3. Fit the 2D-Gabors to the the receptive field estimates

We leave the details of (2) and (3) aside for an extended abstract and refer the reader to Ringach (2002) and Zylberberg et al. (2011). For step (1):

**Encoder:**

Let $\mathbf{A}_{ij}^{(0)} \sim \mathcal{N}(0,1)$ and $\mathbf{x}^{(0)} = \tilde{\mathbf{x}}^{(0)} = \mathbf{0}$

Then, given $\mathbf{A}$ and $\mathbf{y}$, we solve for $\mathbf{x}^* \in \operatorname{argmin}_{\mathbf{x}} \mathcal{L}(\mathbf{A}, \mathbf{y}, \mathbf{x})$ by projected gradient descent:

$$\mathbf{x}^{(t+1)} = \mathcal{P}_S \left( \tilde{\mathbf{x}}^{(t)} - \alpha \nabla_{\mathbf{x}} \mathcal{L}(\mathbf{A}, \mathbf{y}, \tilde{\mathbf{x}}^{(t)}) \right) \tag{5}$$

$$\tilde{\mathbf{x}}^{(t+1)} = \mathbf{x}^{(t+1)} + \gamma^{(t)}(\mathbf{x}^{(t+1)} - \mathbf{x}^{(t)}) \tag{6}$$

for $t \in [T]$. In the code, we run $T = 15$ iterations of projected gradient descent (similar to FISTA). We have $\alpha = \sigma_{\max}(\mathbf{A})^{-2}$ and $\gamma^{(t)}$ is given by:

$$\gamma^{(t)} = \frac{\eta^{(t)} - 1}{\eta^{(t+1)}}, \qquad \eta^{(t+1)} = \frac{1 + \sqrt{1 + 4\eta^{(t)}}}{2}, \qquad \eta^{(0)} = 0 \tag{7}$$

The gradient of the weighted-$\ell_1$ penalty is given by:

$$\nabla_{\mathbf{x}} \mathcal{L}^{WL}(\mathbf{A}, \mathbf{y}, \mathbf{x}) = \mathbf{A}^T(\mathbf{A}\mathbf{x} - \mathbf{y}) + \lambda \sum_{i=1}^{m} ||\mathbf{y} - \mathbf{a}_j||^2 \mathbf{e}_j \tag{8}$$

We also explored a Laplacian penalty $S_\lambda^{LAP}(\mathbf{X}) = \operatorname{tr}(\mathbf{X}\mathcal{G}\mathbf{X}^{\mathbf{T}})$ to promote locality. The gradient of this penalty is most clean when written in a batch setting:

$$\nabla_{\mathbf{X} \in \mathbb{R}^{m \times b}} \mathcal{L}^{LAP}(\mathbf{A}, \mathbf{Y}, \mathbf{X}) = \mathbf{A}^T(\mathbf{A}\mathbf{X} - \mathbf{Y}) + \lambda \left( \mathbf{I}_{b \times b} + \mathbf{X}(\mathcal{G}^T + \mathcal{G}) \right) \tag{9}$$

where $D - A = \mathcal{G} \in \mathbb{R}^{b \times b}$ is a graph Laplacian built from a binary $k$NN graph on the inputs $\mathbf{Y}_{n \times b}$; that is, the edge weight between $\mathbf{y}_i$ and $\mathbf{y}_j$ is 1 if $i, j$ are $k$-nearest neighbors and 0 otherwise. We choose $k = 4$ in our experiments, though more rigorous analysis is required to determine the effect of this hyperparameter.

**Decoder:**

The decoder is a simple linear readout, where given $\mathbf{A}$ and $\mathbf{x}$, $\hat{\mathbf{y}} = \mathbf{A}\mathbf{x}$

## Appendix B. Discriminative Task on Whole Images

Much to our surprise, the weighted-$\ell_1$ loss and unrolled architecture also seems to learn Gabor-like filters even on whole (albeit small) images in addition to random image patches. Below, we include a set of filters learned on CIFAR10, for example. Although these filters are less clean than those learned on the original Sparsenet data (Olshausen and Field, 1997), this offers a path to training an end-to-end discriminative classification task in the spirit of (Rolfe and LeCunn, 2013). How do the categorical and part of units of that paper align with the well-tuned and broadly-tuned cells of the visual cortex, if at all?

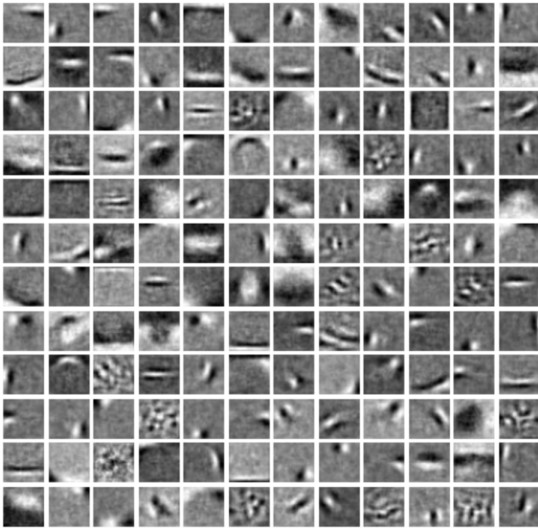

Figure 6: Learned filters when training the weighted-$\ell_1$ loss on CIFAR10. While these images are $32 \times 32$, the receptive fields appear to be quite localized in their sensitivity.

