# OpenReview forum: "Sparse, Geometric Autoencoder Models of V1"
_NeurIPS.cc/2022/Workshop/NeurReps — NeurReps 2022 Poster_

### Official Review · Reviewer_5Xmp · 2022-10-07
**More discussion of method?**

**Confidence:** 3
**Soundness:** 2
**Presentation:** 2
**Contribution:** 2
**Overall Rating:** 5

**Summary:**

The authors extend the sparse coding framework of Olshausen and Field to use a different 'weighted' l1 regularization objective. They show that the resulting objective drives learning of basis functions which are more similar to V1 receptive fields in their degree of tuning.

**Questions:**

How good is the weighted l1 loss for reconstruction compared to the original sparse coding framework? Is there a price to pay in accuracy for the improved match to the Gabor-like V1 filters?

It is not super clear to me why 'algorithm unrolling' is used to learn 'A' and 'X' in Equation 1 -- is this more efficient than simple gradient descent on the objective in equation 1?

If I understand the method correctly, it is really a linear model (i.e. Yhat = AX is linear in X) and the 'recurrence' is only used for optimization. It therefore feels somewhat misleading to call it a 'deep recurrent autoencoder', and I would suggest instead focusing some of the discussion on the properties of different types of regularization rather than 'deep' vs 'shallow' models.

It is not clear to me what the utility is of discussing the iterative Laplacian scheme in equation (4) - is this used anywhere in the abstract?

**Limitations:**

It would be good to explore how their method compares to classical sparse coding on other metrics than just the distribution of spatial phases.

**Recommended Decision:**

2: Borderline

**Relevance:**

3: Solid fit

**Strengths And Weaknesses:**

Expanding theories of sparse coding to better capture the features of biological circuits is an area of much interest in neuroscience, and the authors provide interesting new ideas about what could be driving the emergence of V1 receptive fields.

A shortcoming of the abstract is that they seem to devote very little space to actually exploring their proposed solution, with the majority of the text and figures focusing on previous work and related methods. This also made it challenging to figure out what exactly their method did, with little description of the architecture (is it just a linear projection?).

**Submission Track:**

Extended Abstract (4 Page)

---

> ### Author Response · Authors · 2022-10-26
> **Thanks for the feedback: many fair points raised**
>
> We likely didn't balance the space constraints in the abstract track well enough in our initial version: our introduction attempted to give the reader ample setup for the problem, as well as an overview of other attacks on this problem. This took up space for our own models. As other reviewers seemed to find this introduction difficult to interpret, we realize we could do better on both fronts.
>
> Your loss point raises an important experimental aspect that we left out but will include: we ran the models for many different choices of $\lambda$ and chose the $\lambda$ with the lowest loss for each. These losses are roughly for the same, especially since so much data is used (100,000 patches were used for both models).
>
> We attempted to give a detailed account of the optimization procedure in the appendix (which I agree could be more explicit). The decoder is a linear projection, but the encoder is highly nonlinear. The unrolled architecture is built in a computational graph in Pytorch, and the nonlinearity is applied at each step of the recurrence. The errors are then backpropagated through time on both A,X. As this is a neural architecture, it is able to be parallelized and runs quite fast, even on a CPU. However, you raise a good point on the notion of depth. Since this is unrolled for at least 15 iterations, the computational graph contains at least as many layers, which is our notion of depth. However, since this is not necessarily central to the ideas presented (even though debatable), we've elected to modify our title.
>
> The iterative Laplacian was mentioned merely to acknowledge that similar ideas have been used in other fields. However, the structure of sparsity that we include explicitly affect both the basis functions and latent representations, and the representations live on the probability simplex.
>
> Thanks again!

---

### Official Review · Reviewer_TLeu · 2022-10-14
**Abstract review**

**Confidence:** 5
**Soundness:** 4
**Presentation:** 4
**Contribution:** 4
**Overall Rating:** 8

**Summary:**

The paper proposes a neural network and training loss for learning a dictionary for natural images. It presents evidence showing that the proposed network learns a more biologically accurate diversity of image filters. They additionally suggest an extension to apply the network to a supervised learning task.

**Questions:**

**Recommendations for future work:**

(1) I recommend that the authors look into [1], which demonstrates that the diversity of filters learned by sparse coding can be considerably increased by increasing overcompleteness. My personal investigations have suggested that this difference in overcompleteness is the reason why Rehn and Sommer learned more diverse RFs, not the alternative loss. Exploring the effect of overcomlpeteness in your own studies would be very compelling, especially given that the advancements in compute capabilities allow for more extensive experiments.

(2)  The distinction between the architecture of the sparse coding network proposed by Rozell et al. and an unrolled autoencoder such as what was presented by Rolfe and LeCun is minimal. I would not suggest leaning on this as a source of novelty in future iterations of the work.

(3) Olshausen & colleagues often note that the reconstruction loss is a poor approximation to the objective of the visual system. It’s encouraging to see more sparse coding work moving away from l2 reconstruction, although I would suggest that you provide a more nuanced summary of the current thinking in the field. For example, as far as I know no one has suggested that explicit image reconstruction is a goal of biological vision systems.

(4) I would recommend that you use the LCA network (Rozell et al.) as your baseline comparison instead of SparseNet. It can be implemented easily in modern neural network libraries, and is more extendable in terms of available sparsity constraints.


(5) In addition to investigating the features learned, I would highly recommend that you explore how _inference_ is different between your proposed network and the SC comparison model. This can be done with typical sparse coding tasks, like denoising or inpainting. You could also explore convergence properties and sparsity vs representation quality.

**References:**
[1] Olshausen, Bruno A. "Highly overcomplete sparse coding." Human vision and electronic imaging XVIII. Vol. 8651. SPIE, 2013

**Limitations:**

If the authors intend to submit a full paper to a machine learning field then they should take time to compare their results to more standard (feed forward) neural network approaches. Convincing the ML field that sparse coding is good for semi-supervised learning tasks is often difficult, and bolstering the work with extensive ablation and comparison experiments will greatly improve the likelihood of acceptance.

**Recommended Decision:**

3: Accept

**Relevance:**

4: Highly relevant

**Strengths And Weaknesses:**

**Originality:**
Overall the work presents an original network and loss combination. The underlying architecture presented in figure 4 is quite reminiscent to that of Rolfe and LeCun, 2003, although their use of the LLD and sparsity losses are different, and in my opinion more compelling, than the standard L2 + L1 approach used in Rolfe and LeCun. I also believe the loss is unique in the sparse coding literature.

**Quality:**
While I would be curious to see several additional experiments (noted below), the methods exposition and experimentation provided are sufficient to support this as a promising direction of research.

**Clarity:**
The paper is well written and the ideas are sufficiently motivated and clearly presented. In a longer format I would appreciate a more pedagogical exposition of the LLD loss and how it compares to the sparse coding loss, which would make the paper more accessible to the broader sparse coding community.

**Significance:**
The work is a promising start for what could be a significant contribution to the field.


**Submission Track:**

Extended Abstract (4 Page)

---

> ### Author Response · Authors · 2022-10-26
> **Thank you for the detailed feedback**
>
> We greatly appreciate the time and attention that the reviewer has given to their suggestions for improvement, especially given their expertise in the field.
>
> (1) and (4) are great suggestions for future experiments. (2) is likely a function of our limited evaluation as hinted at in (5) and in the reviews of other authors: clustering is at the core of the latent representations given the graph Laplacian interpretation, and we could've made this clearer with more results. In an extended abstract, we mainly wanted to convey the promise of our initial results and what future experiments might look like, but agree with the point made in (2) overall.
>
> Our paper could certainly benefit from a deeper dive into (3). At an intuitive level, we do not expect the l2 objective to capture the objective of the visual system, either. We look at Gabors as the first-order approximation: it at least gives us some notion that additional structure to that sparsity can improve filters. We will dig deeper into other works that avoid l2 as we expand to more detailed analyses.
>
> Thanks again!

---

### Official Review · Reviewer_qRhc · 2022-10-18
**A bit hard to interpret**

**Confidence:** 2
**Soundness:** 2
**Presentation:** 2
**Contribution:** 2
**Overall Rating:** 3

**Summary:**

The paper is proposing to use deep recurrent autoencoders to neural science.

**Questions:**

- In the previous work, can the authors formulate the problem, i.e. the goal, the challenge, the meaning of the variables? The current version seems hard to interpret mathematically.
- How should we interpret figure 3? Specifically, what does the height of the histograms mean?
- Why is the propose framework in section 3 superior to the previous works?

**Limitations:**

See the weakness and strengths section.

**Recommended Decision:**

1: Reject

**Relevance:**

2: Limited relevance

**Strengths And Weaknesses:**

Weakness:
- Unfortunately, the presentation of the current version of the paper is not friendly for readers that are not familiar with the field.

**Submission Track:**

Extended Abstract (4 Page)

---

> ### Author Response · Authors · 2022-10-27
> **Clarity issues**
>
> We thank the reviewer for their feedback and have attempted to add more explanation for the setup in the camera-ready version with the benefit of an additional page.

---

### Decision · Program_Chairs · 2022-10-21

Accept (Poster)